# Spatial pattern and determinants of sufficient knowledge of mother to child transmission of HIV and its prevention among Nigerian women

Paul Omoh Olopha⬥*, Akin Olusoga Fasoranbaku, Ezra Gayawan

Department of Statistics, The Federal University of Technology, Akure, Nigeria

* poolopha@futa.edu.ng

⬣ OPEN ACCESS

**Data Availability Statement:** The data underlying the results presented in the study are available from the National Population Commission, Nigeria,

## Abstract

The lack of sufficient knowledge of mother to child transmission (MTCT) of human immuno-deficiency virus (HIV) among pregnant women is considered a major contributor to new pediatric HIV infections globally, and increasing HIV related infant mortality especially in developing countries. Nigeria has the highest number of new HIV infections among children in the world. This study was designed to examine the spatial pattern and determinants of acquisition of sufficient knowledge of MTCT and prevention of mother to child transmission (PMTCT) in Nigeria. The data used in the study were extracted from the 2018 Nigeria Democratic Health Survey. The spatial modeling was through a Bayesian approach with appropriate prior distributions assigned to the different parameters of the model and inference was through the integrated nested Laplace approximation technique (INLA). Results show considerable spatial variability in the acquisition of sufficient knowledge of MTCT and its prevention with women in the southwestern and southeastern part of the country having higher likelihood. The nonlinear effects findings show that acquisition of sufficient knowledge of MTCT and PMTCT increased with age of women and peaked at around age 35yearswhere it thereafter dropped drastically among the older women. Furthermore, sufficient knowledge of MTCT and PMTCT was found to be driven by ethnicity, respondents' education and wealth status.

## 1. Introduction

Globally, sub-Saharan Africa accounts for the largest burden of human immunodeficiency virus (HIV) among the adults and children [1]. Specifically, out of the 39 million people living with HIV worldwide [2], about 67% reside in sub-Saharan Africa; 41% are women aged 15 years and older, and approximately 80% of these women live in sub-Saharan Africa [3]. Nigeria is second to South Africa in the number of people living with HIV/AIDS worldwide, representing 9% of the global burden of the disease [4].

In recent years, the various efforts and strategies put in place by different international and national bodies in the fight against the continued spread of HIV has witnessed a significant

ICF International. Nigeria demographic and health survey 2018 (https://dhsprogram.com/pubs/pdf/FR293/FR293.pdf).

**Funding:** The author(s) received no specific funding for this work.

**Competing interests:** The authors have declared that no competing interests exist.

decline in the incidence of morbidity and mortality occasioned by the infection globally [5]. However, published reports have shown that this positive development is being threatened by the continued rise in the number of new infections [5, 6], much of which can be traced to sub-Saharan Africa. In 2017, according to the joint United Nations program on HIV/AIDS (UNAIDS) estimates, 159,000 of the 180,000 new infections among children globally occurred in sub-Saharan Africa, and Nigeria alone accounted for 23% of these new infections in the sub-region. Though Nigeria was able to record a 26% decrease in the number of AIDS-related deaths between 2010 and 2017 from 72,000 to 53,000, the number of new HIV infections rose, however, from 120,000 to 130,000 in the same period [7]. According to the National Agency for the Control of AIDS (NACA), Nigeria has more HIV-infected babies than anywhere in the world [8, 9].

Studies have shown that the major medium of transmission of HIV virus from mother to child has been through pregnancy, delivery and breastfeeding, accounting for 90% of cases [1, 4, 10–13]. This is referred to as mother-to-child transmission of HIV (MTCT), also known as vertical transmission of HIV. Several countries, including Nigeria have made remarkable effort towards elimination of vertical transmission of HIV [5, 14] through the use of antiretroviral therapy (ART) and prevention of mother to child transmission (PMTCT) programmes. Nigeria introduced PMTCT in 2001 but the poor performance of the programme when compared with other countries, some of which are less economically viable, has been of great concern to local authorities [8]. For example, despite its scaled–up response for PMTCT over the years, Nigeria still contributes the greatest number of infants infected with HIV worldwide [8, 12, 15].The reason has been identified to be low accessibility of PMTCT services which has made it difficult to identify pregnant women with HIV and thus exposing the unborn babies to HIV infections before and after delivery. A major gap identified by various reports for the low uptake of PMTCT services has been insufficient knowledge of MTCT and its prevention [16]. The information pregnant women receive about MTCT and PMTCT play an important role in access and utilization of PMTCT services. Sufficient knowledge of MTCT and PMTCT is expected to engender positive attitude by pregnant women towards PMTCT services. In a study undertaken in two health facilities in eastern Nigeria, a significant association was reported to exist between knowledge and utilization of PMTCT services among HIV positive pregnant women [16]. It has also been established that knowledge of MTCT and its prevention is correlated with the desire to have knowledge of one's HIV status and to seek for an HIV testing center [17, 18]. Given this scenario, there is the need for an evidence based study that examines the driving factors of acquisition of sufficient knowledge of MTCT and PMTCT among pregnant women in Nigeria. Most health indicators including health-seeking behaviour by women in Nigeria have been reported to vary substantially across geographical locations [19]. Consequently, this pattern of variation could impact on the women's knowledge of available health services particularly those related to HIV preventive measures, and this needs to be taken into account when analyzing women's knowledge of MTCT and PMTCT. Such study would be of great benefit in identifying areas of low and high risks which will, in turn, enhance intervention efforts and implementation of relevant policy strategies in eradicating MTCT and significantly reducing or even eliminating the HIV pandemic by the year 2030 as conceived by the Sustainable Development Goal 3.3 (SDG 3.3). It is pertinent to note that studies conducted on this issue have been limited to understanding the associated determinants in different parts of the country. To our knowledge, there has been no study that examines the spatial patterns of women's knowledge of MTCT and PMTCT in Nigeria. Consequently, this study was designed to analyse the spatial distributions of the knowledge of MTCT and PMTCT among women of reproductive age in Nigeria employing robust techniques that can accommodate variables of different types. In this regard, the study has appealed to the use of structural additive regression method where both the functional forms of metrical variables and spatial parts can be simultaneously combined and treated as component variables of a single equation. Findings

from the study would aid understanding of locations across Nigeria where the knowledge of these important HIV prevention initiatives are limited among these women.

## 2. Methods

### 2.1 Study design and setting

The data used for the study were extracted from the Nigeria Demographic and Health Survey (NDHS) conducted in 2018. The data source is considered appropriate for the study because the observations are geographically referenced that permits spatial modeling. The sampling frame used for the survey was created using that of the 2006 Nigerian Population and Housing Census. The sampling units were defined based on the enumeration areas (EAs) used for the census frame. A stratified two-stage sampling design was used to select the sample. The first stage sample included 1400 clusters (sampling units) made up of 577 in urban areas and 813 in rural areas from where a total of 41,821 households were selected. A total of40,427 women were found eligible for interview in the households and there was a 96.7% response rate. The data analysed came from 34,837 women of reproductive age who have complete information on the five relevant questions from which information on MTCT and PMTCT were derived.

### 2.2 Outcome variable

Similar to [5, 20], the outcome variable in this study was considered to be sufficient knowledge of MTCT and PMTCT. To arrive at this, five relevant questions were asked during the survey namely: (i) "Have you ever heard of HIV or AIDS?" (ii) "Can HIV be transmitted from a mother to her baby during pregnancy?" (iii) "Can HIV be transmitted from a mother to her baby during delivery?" (iv) "Can HIV be transmitted from a mother to her baby during breastfeeding?" (v) "Are you aware of drugs to avoid HIV transmission to baby during pregnancy?" Responses to each of the questions were coded as 1 if responded gave a yes answer and 0 if no. A respondent with a total score of 5 was regarded as having sufficient knowledge of MTCT and PMTCT while a respondent with less than a total score of 5 was regarded as having insufficient knowledge of MTCT and PMTCT. A binary composite variable was therefore created based on the scores.

### 2.3 The independent variables

The individual level covariates were demographic variables such as respondent's current age, place of residence, state (district), marital status, religion, highest educational level attained, ethnicity, wealth index and level of access to mass media (frequency of reading newspaper, listening to radio and watching television).

Table 1 presents the descriptive statistics of all women included in the study. The mean age of the 34,837 respondents was 29.6 years, out of which 17.9% were less than 20years old, 34.2% between 20 and 29years, 28.4% were between 30 and 39 years and 19.6% were 40 years and above. Majority of the respondents reside in rural areas (58.0%) and about equal proportions of the respondents belong to the two major religions, Christianity (51.3%) and Islam (47.9%), practiced in the country. Furthermore, the three major ethnic groups in Nigeria were represented in the study population as follows: Hausa/Fulani (31.9%), Igbo (17.1%) and Yoruba (12.8%). All other minority tribes in the country totaling about 250 were grouped together to form the remaining 38.2%. Majority of the respondents were in union (75.9%), and currently working at the time of the survey (66.3%). However, a high percentage of the women were either not educated (31.9%) or completed secondary school (41.4%). The proportions of respondents in the different wealth quintiles were almost equal though those in the middle and

**Table 1. Summary of individual covariates analysed.**

| Variables | All women respondents | |
|---|---|---|
| | Number of respondents | Percentage (%) |
| **Place of Residence** | | |
| Urban | 14625 | 42.0 |
| Rural | 20212 | 58.0 |
| **Current marital status** | | |
| Never married | 8409 | 24.1 |
| Ever married | 26427 | 75.9 |
| **Religion** | | |
| Others | 260 | 0.8 |
| Christians | 17879 | 51.3 |
| Islam | 16698 | 47.9 |
| **Ethnicity** | | |
| Others | 13301 | 38.2 |
| Hausa/Fulani | 11097 | 31.9 |
| Igbo | 5970 | 17.1 |
| Yoruba | 4469 | 12.8 |
| **Woman's highest educational level** | | |
| No education | 11134 | 32.0 |
| Primary | 5289 | 15.2 |
| Secondary | 14417 | 41.3 |
| Higher | 3997 | 11.5 |
| **Wealth Index** | | |
| Poorest | 5914 | 17.0 |
| Poorer | 6650 | 19.1 |
| Middle | 7433 | 21.3 |
| Richer | 7742 | 22.2 |
| Richest | 7098 | 20.4 |
| **Respondent's working status** | | |
| Currently working | 23104 | 66.3 |
| Not currently working | 11733 | 33.7 |
| **Access to mass media** | | |
| **Newspaper** | | |
| Yes | 5744 | 16.5 |
| No | 29122 | 83.5 |
| **Radio** | | |
| Yes | 20058 | 57.6 |
| No | 14779 | 42.4 |
| **Television** | | |
| Yes | 18338 | 52.6 |
| No | 16499 | 47.4 |
| Talked about testing for HIV during antenatal visit | | |
| Yes | 9384 | 77.2 |
| No | 2766 | 22.8 |
| Woman's age at the time of survey | A continuous variable measured in year | |
| Woman's age at marriage (ever-married women) | A continuous variable measured in year | |

richer quintiles were slightly higher (21.3% and 22.2%) respectively. Half of the women watched television at least once in a week, radio (57.6%), and newspaper, a dismal 16.5%.

Table 2 presents the summary of the proportions of women with sufficient knowledge of MTCT and PMTCT. The Table shows that 100% of the respondents have heard of HIV and AIDS, out which 77.2% have the knowledge of MTCT during pregnancy, 82.3% during delivery and 94.3% during breastfeeding. Furthermore, 85.3% alluded to knowledge of the existence of drugs to prevent MTCT of HIV. Consequently, a composite score of the five variables that constitute sufficient knowledge of MTCT and PMTCT in this national study reveals that 59.2% of the women have the sufficient knowledge in Nigeria. An earlier study held in a health facility in a section of the country by [21] put the percentage of women with sufficient knowledge of MTCT and PMTCT at 78%.

## 2.4 Statistical analysis

Let $y_i$, $i = 1, \ldots, n$, be a random variable indicating whether or not a pregnant woman $i$ has sufficient knowledge of MTCT and PMTCT. Then $y_i$ is said to follow a binomial distribution with parameter $p_i$, the probability of success, where $p_i$ is taken to be the proportion of pregnant women with sufficient knowledge of MTCT and PMTCT. If a set of explanatory variables denoted by $x_i$, $i = 1, \ldots, k$ are associated with $y_i$, then the binary response variable can be modeled conditional on the covariates by using a binary logistic regression [22]. The logistic model is given by

$$g(p_i) = \log\left(\frac{p_i}{1 - p_i}\right) = \eta_i = \beta_0 + \beta_1 x_1 + \ldots + \beta_k x_k \tag{1}$$

where $\eta_i$ is the model predictor, $\beta_0$ is the regression constant term, and $\beta_k$ is the regression coefficient for the k-th explanatory variable. Two metrical variables that are relevant to the

**Table 2. Knowledge of MTCT and PMTCT among women (15–49 years) in Nigeria in 2018 (N = 34837).**

| Knowledge of MTCT and PMTCT | Number of respondents | Percentage |
|---|---|---|
| | | |
| **Ever heard HIV and AIDS** | | |
| No | 1 | 0 |
| Yes | 34836 | 100.0 |
| **HIV transmitted during pregnancy** | | |
| No | 7950 | 22.8 |
| Yes | 26887 | 77.2 |
| **HIV transmitted during delivery** | | |
| No | 6166 | 17.7 |
| Yes | 28671 | 82.3 |
| **HIV transmitted during breastfeeding** | | |
| No | 1993 | 5.7 |
| Yes | 32844 | 94.3 |
| **There are drugs to avoid HIV transmission to the child during pregnancy** | | |
| No | 5138 | 14.7 |
| Yes | 29699 | 85.3 |
| **Knowledge of MTCT and PMTCT** | | |
| Insufficient | 14226 | 40.8 |
| Sufficient | 20611 | 59.2 |

study, respondent's age and age at the time of first birth ($z_1$ and $z_2$) were included as explanatory variables. We therefore appeal to the use of non-parametric approach to take care of the effects of the metrical variables as they cannot be described in simple functional forms. The geo-referenced location of residence of the respondents was represented in a discrete form, $s_i$, and used in the analysis. Based on the instrumentality of the concept of structural additive regression, the logistic model in Eq (1) is then extended to capture the different types of variables at once through the structural additive model defined as

$$\log\left(\frac{p_i}{1 - p_i}\right) = \sum_{j=1}^{p} f_j(z_{ij}) + f_s(s_i) + x_i\beta \tag{2}$$

where $f_1, \ldots f_p$ represent the nonlinear functions considered for the metrical variable and $f_s$, the nonlinear spatial effect of the discrete spatial covariates.

The common approach of estimating the parameters of a structured additive model is the Bayesian approach, where the parameters and functions are considered to be random variables and prior distributions are assigned to them. For the linear parameters, we assumed a vague normal prior with mean zero and small precision $\tau^2$. The nonlinear effects of the metrical covariates were modeled by assigning a second order Gaussian random walk described as

$$\beta_t = 2\beta_{j-1} - \beta_{j-2} + u_t,$$

where $u_t$ is iid and $N(0, \tau^2)$ with the variance, $\tau^2$ controlling the smoothness of the B-spline function. A weakly informative inverse Gamma prior, $\tau^2 \sim IG(a, b)$ is assigned to the variance. For the spatial component, a common approach in spatial statistics is to adopt a Gaussian Markov random field prior [23]. It is given as

$$[f_s(s)|f_s(t); t \neq s, \tau^2] \sim N\left[\sum_{t\in\partial_s} \frac{f_s(t)}{N_s}, \frac{\tau^2}{N_s}\right],$$

where $N_s$ is the number of adjacent regions and $t\in\partial_s$ denotes that region $t$ is a neighbor of other regions.

The posterior distribution for model of this nature cannot be handled analytically, consequently, inference in this study was based on Integrated Nested Laplace Approximations (INLA) as implemented in R-INLA, a packaged built within the R statistical package for approximating complex Bayesian models [24]. Further details on structured additive regression models and other different parameters that can be accommodated including the estimation procedure can be found in Fahrmeir et al. [22].

## 2.5 Model building

To analyze the data, three models of different specifications, starting from the simplest to the more complex ones were examined. The basic or primary model (M1) included only the geographical areas as random effects. The aim was to determine the crude variations in women acquisition of sufficient knowledge of MTCT and its prevention among the states of Nigeria without adjusting for any covariates. Subsequent addition of covariates helps us to understand their effects on the response variable. The second model (M2) was fitted adjusting for the covariates and the non-linear effects. Finally, random effects at household and community levels were included to take care of unobserved heterogeneity that may be present in the data. The various models explored are the following:

Model $M_1 \eta_{ijkl} = S_i$

Model $M_2 \eta_{ijkl} = M_1 + x_{ijkl}\gamma + f(\text{current age of woman}) + f(\text{number of ANC visits}) + f(\text{woman's age at marriage})$

**Table 3. Summary of model diagnostic criteria.**

| Model | Model Diagnostic | All women of reproductive age | Ever-married women |
|-------|------------------|-------------------------------|--------------------|
| M1    | DIC              | 43706.92                      |                    |
| M2    | DIC              | 43440.04                      | 20926.58           |
| M3    | DIC              | 42113.19                      | 20314.44           |

Model $M_3 \eta_{ijkl} = M_2 + \mu_{ijk} + \nu_{ij}$ where the effects are fixed ($\gamma$), random ($\mu_{ijk}$, $\nu_{ij}$), non-linear (f) and spatial ($S_i$)

The data were first analysed for all women respondents (ever-married and never-married) by using Models $M_1$–$M_3$. The analyses were then repeated for ever-married respondents who reported to have either attended ANC or not during pregnancy. This was to enable us measure the effect of marital and partner-related covariates on the acquisition of knowledge of MTCT and PMTCT. This is because never-married women did not have partners for which data could be collected. Model diagnostic was based on deviance information criterion (DIC). The model regarded as the best is the one with the lowest DIC value [25].

The model diagnostic statistics for all the models examined are presented in Table 3. The table presents the results for all women respondents and ever-married respondents separately. For both data sets, the DIC for model M3 has the smallest value, revealing that controlling for random effects at both household and community levels in a single model provided the best fit. Discussion of results is therefore based on Models M1 and M3.

## 3. Results

Fig 1 presents the spatial effects of the pregnant women with the sufficient knowledge of MTCT of HIV and its prevention. Parts (a) and (c) of the figure present the maps of posterior means for all women respondents while (b) and (d) show the maps of the location of the 95% credible intervals respectively. Also, part (e) shows the map of posterior means for ever-married women respondents while (f) shows the map of the location of the 95% credible intervals. The credible intervals were used in deciding the significance of the posterior mean estimates. From the maps of credible intervals, states in purple (red) shading are places where the estimates for respondents' sufficient knowledge of MTCT and PMTCT are significantly higher (lower) whereas, estimates for states shaded in white colour are not significant. The geographical pattern of the crude probability of acquisition of sufficient knowledge of MTCT and PMTCT (Fig 1A and 1B) reveals a strong spatial inequality among the states with those in the south-west and most parts of South-east having significantly higher chances of acquiring the required knowledge, whereas most states located in the south-south and North-eastern parts of the country have significantly lower chances of acquiring the required knowledge. However, estimates for Sokoto, Katsina, Niger Rivers and Lagos states are not significant. There is, however, a slight change in the pattern after adjusting for covariates and unobserved heterogeneity. The net effect reveals that respondents living in the southwestern part of Nigeria: Oyo, Ogun, Ekiti, and Osun states excepting Ondo state were significantly more likely to have acquired sufficient knowledge of MTCT and PMTCT while the those resident in the whole of North Eastern region specifically Borno, Bauchi, Yobe, Adamawa, Taraba, and Gombe states were significantly less likely to have sufficient knowledge of MTCT and PMTCT. Furthermore, in the South-south region, prevalence of sufficient MTCT and PMTCT knowledge was significantly higher among pregnant women in Bayelsa state only, but significantly lower in Edo and Delta states while estimate for Rivers, Cross-River, and Akwa-Ibom states are not significant. The southeastern states of Enugu, Ebonyi, and Abia are associated with pregnant women who

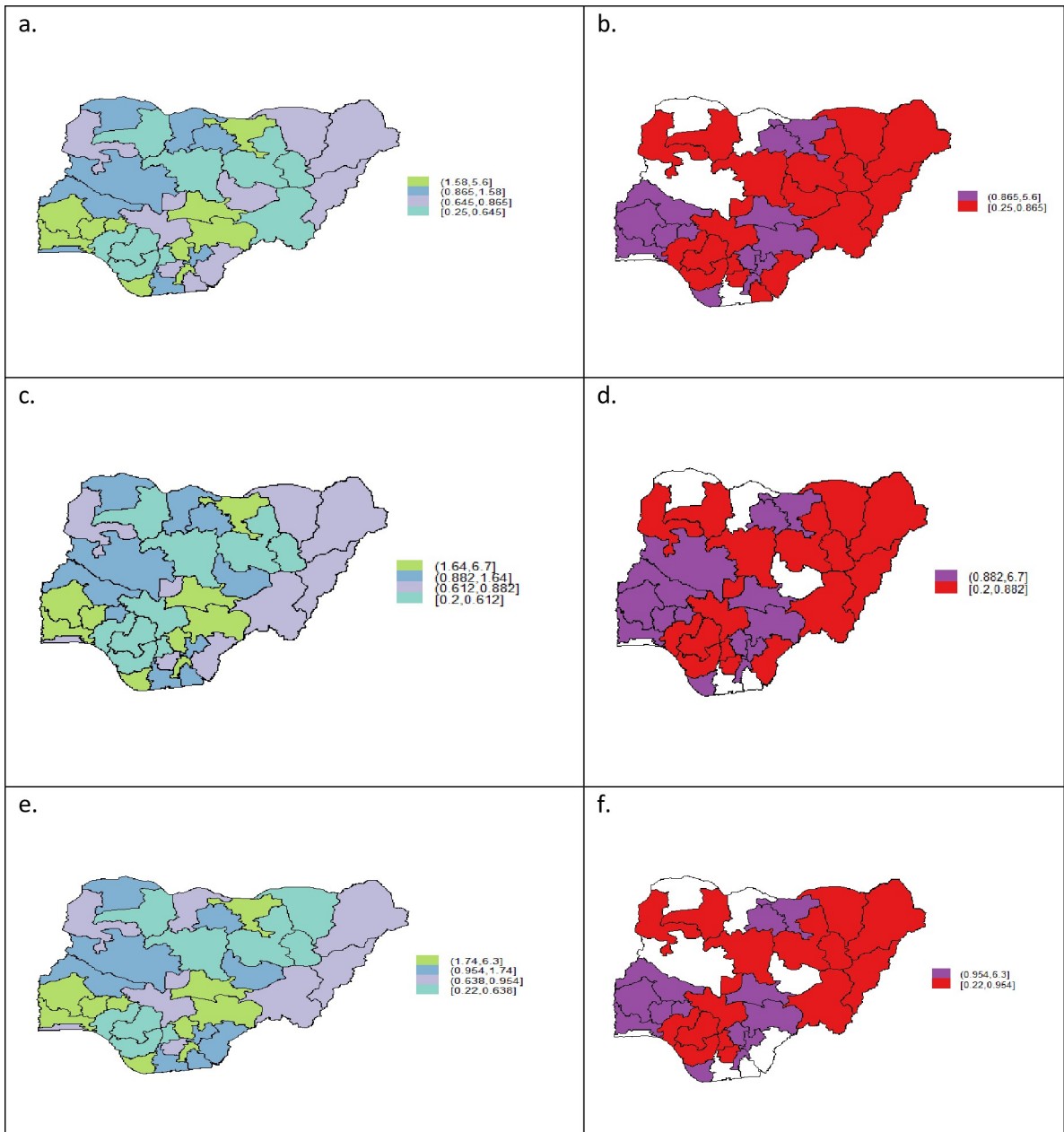

**Fig 1.** Estimated posterior mean spatial effects of Models (a) M1 and (b) its corresponding 95% posterior probability; (c) M3 for all women respondents and (d) its 95% posterior probability; and M3 for ever-married respondents and (f) its 95% posterior probability.

have higher likelihood of sufficient knowledge of MTCT and PMTCT while those resident in Anambra and Imo states had lower chances. The only states in the core northern Nigeria where pregnant women were more likely to possess sufficient knowledge of MTCT and PMTCT were Kano and Jigawa located in the North-West region. Pregnant women in all the other states in the Northwest were either less likely or the estimates are not significant.

Spatial effects for the ever married respondents in (Fig 2E and 2F) show that acquisition of sufficient knowledge of MTCT and its prevention is highly prevalent among women in the same states as that of all women respondents except in Niger state where there was no

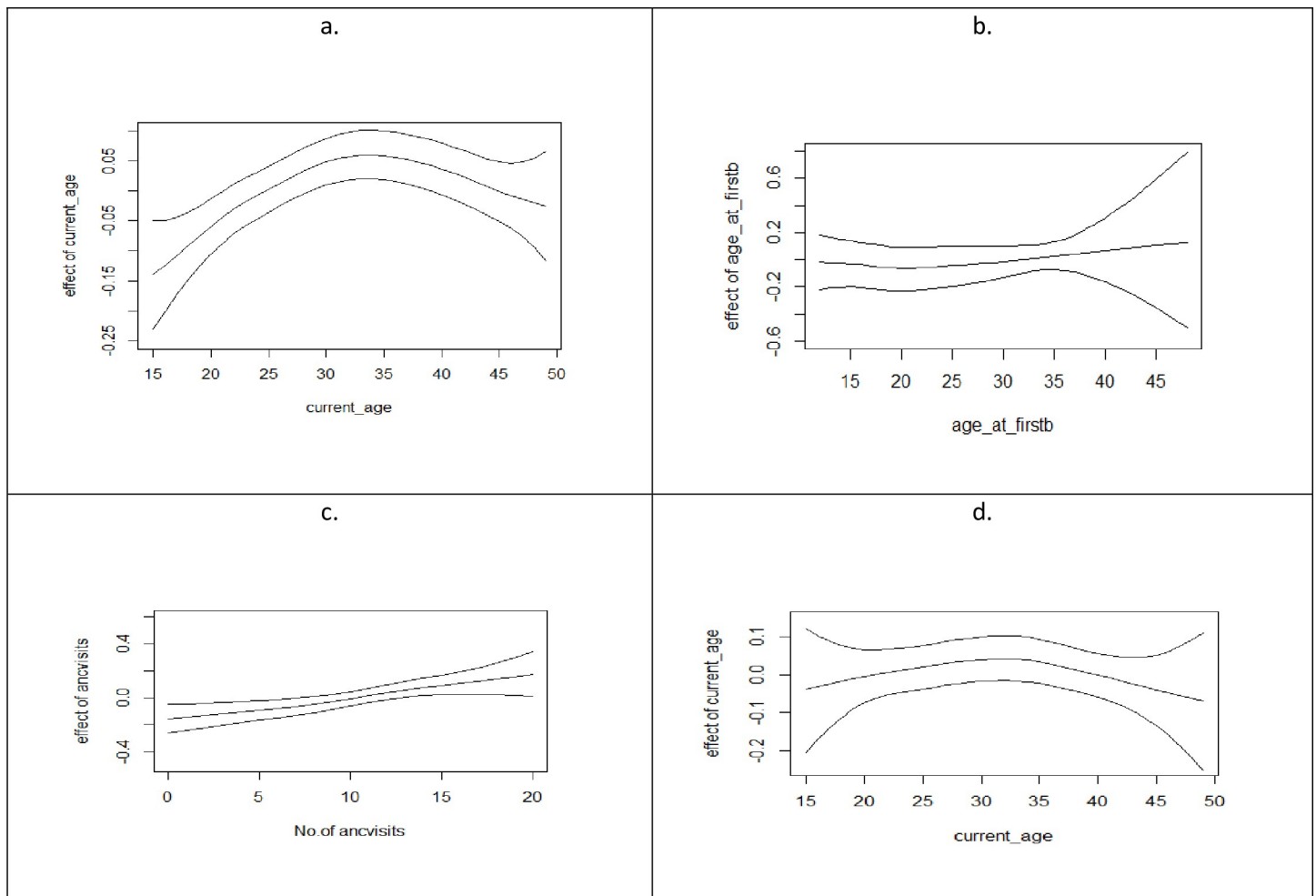

**Fig 2.** Non-linear effects of (a) woman's current age at the time of survey, (b) age at first birth, (c) number of antenatal visits, and (d) woman's current age at the time of survey (ever-married women).

significant association. Similarly, acquisition of sufficient knowledge of MTCT and PMTCT was significantly lower in the same states as that of all women respondents except in the South-south state of Cross River where there was non-significant association.

Estimates for the nonlinear effects of the pregnant women's current age as at the time of the survey, woman's age at marriage, and the number of antenatal visits during pregnancy, on the acquisition of sufficient knowledge of MTCT and PMTCT, are presented in (Fig 2A–2D). The posterior means are represented by the middle lines, flanked by the 95% credible intervals.

Results show similar pattern for current age of women at the time of the survey for both all women (Fig 1A) and ever-married women (Fig 1D) respondents. Acquisition of sufficient knowledge of MTCT and PMTCT increased for every unit increase in age until around age 35years from where it reduced drastically among the older women. This shows that in Nigeria, younger women under 35 years of age have more chances of being conversant with the knowledge of MTCT and its prevention than the older women. Fig 1B however shows acquisition of sufficient knowledge of MTCT and PMTCT remained at zero level among pregnant women whose age at first birth was below 20 years. The pattern thereafter changed considerably, revealing a direct linear relationship between age at first birth (from 20 years and above) and

acquisition of sufficient knowledge of MTCT and PMTCT. This information signifies that the older women (women who were advanced in age) at the time of first birth possess more chances of acquiring sufficient knowledge of MTCT and PMTCT than those who start child bearing at younger ages. Acquisition of sufficient knowledge of MTCT and PMTCT was found to be directly related to the number of antenatal visits made by pregnant women (Fig 1C), indicating that the likelihood of acquiring sufficient knowledge of MTCT and PMTCT increases for every unit increase in the number of ANC visits.

Table 4 shows the posterior odds ratio and 95% credible intervals for the determinants of sufficient knowledge of MTCT and PMTCT. The results show that the likelihood of pregnant women in Nigeria having sufficient knowledge of MTCT and PMTCT was significantly associated with ethnicity, respondents' education, and wealth status. Specifically, the study revealed that pregnant women who were of Yoruba ethnic extraction were more likely to possess sufficient knowledge of MTCT and PMTCT compared with pregnant women from other ethnic groups. Pregnant women who attained secondary or higher level of education had higher odds of sufficient MTCT and PMTCT. Likewise, the results show that women in the richest class of the society have significantly higher odds for acquiring sufficient knowledge of MTCT and PMTCT. On the other hand, women who were never married were less likely to have sufficient knowledge of MTCT and its prevention when compared with their ever-married counterparts. Similarly, women who watched television or listen to radio at least once a week had significantly lower odds of having sufficient MTCT and PMTCT. In addition, this study found no association between sufficient knowledge of MTCT and PMTCT and the two major religions practiced in Nigeria (Christianity and Islam) and place of residence (whether rural or urban). The average probability of a woman of reproductive age in Nigeria acquiring sufficient knowledge of MTCT and PMTCT was estimated to be 0.7.

Results for ever-married respondents presented in Table 4 show that when compared with all women respondents, ever-married women who had partners with academic attainment not more than primary education had significantly lower likelihood of acquiring sufficient knowledge of PMTCT and PMTCT. Findings for other variables are similar to those of all women respondents. However, results show that though ever-married women in the richest class of the society also have higher odds for acquiring sufficient knowledge of MTCT and PMTCT, it was not significant.

## 4. Discussion

The spatial pattern and determinants of sufficient knowledge of mother to child transmission and prevention has been analysed out using a structured additive modeling approach. The results show a strong spatial variability that exists among the various states of Nigeria. Pregnant women with sufficient MTCT and its prevention were shown to be more prevalent in the southern states than in the northern states with the states located in the northeast being the greatest culprit. This could be attributed to the fact that the women from southern Nigeria are more educated and economically advantaged than those in the north. Economic empowerment or wealth and educational attainment of the respondents have been shown in this study to constitute the major driving factors for sufficient knowledge of MTCT and its prevention in Nigeria [see Table 2]. Furthermore, the spatial distribution of states with sufficient knowledge of MTCT and its prevention tends to reflect the pattern followed by the spatial distribution of antenatal care utilization by pregnant women in Nigeria as reported by [19]. This reflection tends to establish various reports of a positive relationship existing between knowledge of MTCT and PMTCT services and attendance at antenatal care facilities [4, 5, 8, 16]. This is further confirmed by Fig 2C which shows that the higher the number of ANC visits a pregnant

**Table 4. Posterior odds ratio for the linear parameters of socio-economic and demographic variables.**

| Variables | All Women | | | Ever-married women | | |
|---|---|---|---|---|---|---|
| | mean | 0.025quant | 0.975quant | mean | 0.025quant | 0.975quant |
| Intercept | 2.242 | 1.627 | 3.115 | 1.197 | 0.636 | 1.769 |
| **Media** | | | | | | |
| Others | 1 | 1 | 1 | 1 | 1 | 1 |
| Newspaper | 0.967 | 0.9 | 1.039 | 0.89 | 0.791 | 1.003 |
| Television | 0.818 | 0.766 | 0.873 | 0.825 | 0.749 | 0.908 |
| Radio | 0.947 | 0.895 | 1.001 | 0.871 | 0.804 | 0.944 |
| **Ethnicity** | | | | | | |
| Others | 1 | 1 | 1 | 1 | 1 | 1 |
| Igbo | 1.052 | 0.919 | 1.203 | 1.2 | 0.969 | 1.487 |
| Yoruba | 1.159 | 1.009 | 1.329 | 1.334 | 1.075 | 1.654 |
| Hausa_ Fulani | 1.043 | 0.955 | 1.141 | 1.082 | 0.962 | 1.219 |
| **Religion** | | | | | | |
| Others | 1 | 1 | 1 | 1 | 1 | 1 |
| Islam | 0.789 | 0.569 | 1.086 | 0.725 | 0.449 | 1.153 |
| Christians | 0.785 | 0.569 | 1.073 | 0.799 | 0.497 | 1.262 |
| **Education** | | | | | | |
| No-education | 1 | 1 | 1 | 1 | 1 | 1 |
| Primary | 1.035 | 0.956 | 1.121 | 0.946 | 0.844 | 1.06 |
| Secondary | 1.084 | 1.001 | 1.173 | 1.015 | 0.899 | 1.146 |
| Higher | 1.371 | 1.228 | 1.531 | 1.26 | 1.044 | 1.52 |
| **Wealth** | | | | | | |
| Poorest | 1 | 1 | 1 | 1 | 1 | 1 |
| Poorer | 1.016 | 0.94 | 1.099 | 1.06 | 0.956 | 1.175 |
| Middle | 0.996 | 0.915 | 1.084 | 1.096 | 0.974 | 1.232 |
| Richer | 1.067 | 0.969 | 1.175 | 1.095 | 0.952 | 1.26 |
| Richest | 1.153 | 1.032 | 1.288 | 1.21 | 1.02 | 1.438 |
| **Place of residence** | | | | | | |
| Rural | 1 | 1 | 1 | 1 | 1 | 1 |
| Urban | 1.061 | 0.999 | 1.126 | 1.077 | 0.984 | 1.177 |
| **Working status** | | | | | | |
| Unemployed | 1 | 1 | 1 | 1 | 1 | 1 |
| Employed | 0.873 | 0.827 | 0.922 | 0.839 | 0.776 | 0.906 |
| **Marital Status** | | | | | | |
| Ever married | 1 | 1 | 1 | - | - | - |
| Never married | 0.799 | 0.741 | 0.86 | - | - | - |
| **Partner/Husband's Education** | | | | | | |
| No Education | - | - | - | 1 | 1 | 1 |
| Primary | - | - | - | 0.86 | 0.765 | 0.967 |
| Secondary | - | - | - | 0.977 | 0.876 | 1.089 |
| Higher | - | - | - | 0.965 | 0.839 | 1.108 |

woman make, the higher the likelihood of acquisition of sufficient knowledge of MTCT and its prevention.

The revelation that the acquisition of sufficient knowledge of MTCT and PMTCT is lower among the older women beyond age 35yearsin Nigeria is consistent with a research finding that showed that older women had lower level of knowledge on PMTCT in Ethiopia [26]. This

could be attributed to the fact that older women may have lesser opportunities for education than the younger ones, which invariably deprive them access to information sources such as newspaper and social media. Furthermore, it could also be explained by earlier research findings that demonstrated women grow older in marriage, their utilization of antenatal care reduces as they tend to rely more on experience in child bearing [19]. These older women may therefore not be conversant or updated with new development in MTCT and PMTCT. The findings therefore suggest that more of the campaign on acquisition of sufficient knowledge of MTCT and PMTCT might not have benefited older women as it did the younger ones. This cannot be unconnected with the reported staggering figures of HIV infected children in Nigeria over the years. There is however hope for the future of maternal and child health in the country with the current positive trend in the likelihood of acquisition of knowledge of MTCT and PMTCT among the younger women as they advance in age. Hence, there is the need for government to sustain the current tempo among the younger women. Nigerian governments at all levels will also do well to urgently step up campaign and effective strategies to educate older women and teenage mother son acquisition of sufficient knowledge of MTCT and PMTCT. All these efforts are very necessary if Nigeria is going to reduce the current staggering number of HIV infections among children considerably [1, 4, 9, 11, 16].

It is interesting to note that contrary to various published articles stating the positive contribution of religion and urban residency to acquisition of sufficient knowledge of MTCT and PMTCT services [5, 27], the two factors have shown no significant contribution to acquisition of sufficient knowledge of MTCT and PMTCT by pregnant women in Nigeria, aligning with a similar earlier report on Nigeria by [21]. The non-significance of religion to acquisition of sufficient knowledge of MTCT and PMTCT establishes earlier similar reports in Nigeria [20, 28–30]. Many researchers have adduced this to the nonchalant attitude of religious groups towards getting involved in women acquisition of knowledge of MTCT and its prevention simply because they perceive immoral behavior as the major cause of the HIV/AIDS epidemic in Africa [28, 31]. On the other hand, the non-significance of urban residency in the acquisition of sufficient knowledge of MTCT and PMTCT in Nigeria contradicts what obtains in various other countries where urban residency was reported to contribute significantly [5]. This could be explained by the result that pregnant women with sufficient knowledge of MTCT and its prevention in this study reside in either rural or urban in almost equal proportion.

Regarding association of respondents' education with sufficient knowledge of MTCT and PMTCT, those women who attained secondary or higher level of education were found to be more likely to have sufficient knowledge of MTCT and PMTCT of HIV compared with those with no education and this might be due to the fact that uneducated women might be different from the educated ones in their understanding of MTCT and PMTCT, and other public health issues. This possible explanation is also in line with the finding from China and Addis Ababa, Ethiopia where women having secondary or higher education were found to have better knowledge of MTCT and PMTCT of HIV than those with no education [11, 32].

## 4.1 Limitations of the study

It is worthy to mention that the analysis carried out in this study did not identify all the variables that may be responsible for the observed spatial structure of the knowledge of MTCT and its prevention in Nigeria. For instances, some cultural values held in parts of Nigeria restrict women from taking independent decisions on healthcare issues, which might in turn limit their attendance at antenatal care services and thus having a strong influence on their ability to acquire the need knowledge of HIV and its prevention. Variables that measure cultural values of this nature are not covered by our data source. Furthermore, the cross-sectional

nature of the data did not give room for causal inference. The spatial units analysed were the states of Nigeria each of which comprises several local government areas hosting people of different languages and culture. This could further cause variations within each state and it would be worthy to unravel these variations in future studies using spatial approaches that consider smaller areas such as adopting a continuous spatial process. The findings were however able to pinpoint states that require urgent intervention in the acquisition of knowledge of MTCT and PMTCT. These findings can lead to development of hypotheses on the underlying reasons for the observed spatial patterns. Further, the results could be useful for planning purposes which is a pivot issue in policy circles that aim at focusing the allocation of resources to the most needed areas.

## 5. Conclusion

In this study, the average probability of a woman of reproductive age in Nigeria acquiring sufficient knowledge of MTCT and PMTCT was computed to be 0.73 and with high spatial variability across the country. It was found to be prevalent among women in states located in the Southwest and Southeast parts of the country. This knowledge is lower among women in the South-south and Northern parts with the Northeastern part of the country been the greatest culprit. The driving factors of acquisition of such knowledge were current age of women at the time of survey (young women), women educational attainment and economic capability, measured by household wealth index. To curb or eliminate further increase in new infections and mortality among children, intervention efforts and campaign should be geared towards increasing the acquisition of knowledge of MTCT and PMTCT among the older women, focus on areas or states with low prevalence of sufficient knowledge, and bringing PMTCT services closer to the poor people, especially in economically disadvantaged locations.

## Author Contributions

**Conceptualization:** Paul Omoh Olopha.

**Data curation:** Paul Omoh Olopha.

**Formal analysis:** Paul Omoh Olopha.

**Investigation:** Paul Omoh Olopha, Akin Olusoga Fasoranbaku, Ezra Gayawan.

**Methodology:** Paul Omoh Olopha, Akin Olusoga Fasoranbaku, Ezra Gayawan.

**Resources:** Paul Omoh Olopha.

**Software:** Paul Omoh Olopha.

**Supervision:** Akin Olusoga Fasoranbaku, Ezra Gayawan.

**Writing – original draft:** Paul Omoh Olopha.

**Writing – review & editing:** Paul Omoh Olopha, Ezra Gayawan.

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
