## [Decision Letter · Decision Letter 0]

5 May 2021

PONE-D-21-02315

Spatial Pattern and Determinants of Sufficient Knowledge of Mother to Child Transmission of HIV and its prevention among Nigerian Women

PLOS ONE

Dear Dr. Olopha,

Thank you for submitting your manuscript to PLOS ONE. After careful consideration, we feel that it has merit but does not fully meet PLOS ONE’s publication criteria as it currently stands. Therefore, we invite you to submit a revised version of the manuscript that addresses the points raised during the review process.

We look forward to receiving your revised manuscript.

Kind regards,

Sara Ornaghi, M.D., Ph.D.

Academic Editor

PLOS ONE

Journal Requirements:

3) For studies involving humans categorized by race/ethnicity, age, disease/disabilities, religion, sex/gender, sexual orientation, or other socially constructed groupings, authors should:

i) Explicitly describe their methods of categorizing human populations,

ii) Define categories in as much detail as the study protocol allows,

iii) Justify their choices of definitions and categories,

iv) Explain whether (and if so, how) they controlled for confounding variables such as socioeconomic status, nutrition, environmental exposures, or similar factors in their analysis

4) Please amend your list of authors on the manuscript title page to ensure that each author is linked to an affiliation. Authors’ affiliations should reflect the institution where the work was done (if authors moved subsequently, you can also list the new affiliation stating “current affiliation:….” as necessary).

5) Please upload a copy of Figure 2, to which you refer in your text (we note you currently have two figures but both are labelled 'Figure 1'.) . If the figure is no longer to be included as part of the submission please remove all reference to it within the text.

Reviewers' comments:

Reviewer's Responses to Questions

**Comments to the Author**

1. Is the manuscript technically sound, and do the data support the conclusions?

Reviewer #1: Yes

Reviewer #2: Yes

2. Has the statistical analysis been performed appropriately and rigorously? 

Reviewer #1: I Don't Know

Reviewer #2: I Don't Know

3. Have the authors made all data underlying the findings in their manuscript fully available?

Reviewer #1: Yes

Reviewer #2: Yes

4. Is the manuscript presented in an intelligible fashion and written in standard English?

Reviewer #1: Yes

Reviewer #2: Yes

5. Review Comments to the Author

Reviewer #1: To me the work is worthwhile to raise awareness of the need to disseminate health and hygiene knowledge among women in order to raise their awareness and prevent new infections. I consider the efforts made by the authors to identify socio-economic determinants that can be the subject of targeted public health interventions to enhance women's awareness of the subject particularly interesting. It is necessary to make a timely review of the spacing, in particular in the "results" section. I am unable to pronounce on the correctness of the statistical analyzes, as they are beyond my knowledge. However, considering the point of view of a reader who is on average fasting in statistical analysis, I believe that applied statistical analysis is expressed in a somewhat verbose way, and not free from some repetitions, ending up making the reading of this section not very smooth. Overall, I think the work is interesting and worthy of being brought to the attention of the public. The data exposed concern many women, thus acquiring considerable weight. The conclusions are in line with the findings and expressed with the necessary emphasis.

Reviewer #2: The article is interesting even if very complicated if you do not have great statistical skills. The methods and or results seem exhaustive. The data in table number 1 should be better specified, in which the characteristics of the general population are described (and not only of the women who then actually answered the survey of the five questions): it is not clear whether it refers to the total of 40,427 women.

6. PLOS authors have the option to publish the peer review history of their article (what does this mean?). If published, this will include your full peer review and any attached files.

Reviewer #1: **Yes: **Francesca Sabbatini

Reviewer #2: No

---

## [Author Response · Author response to Decision Letter 0]

1 Jun 2021

RESPONSES TO REVIEWERS’ COMMENTS

PONE-D-21-02315

Spatial Pattern and Determinants of Sufficient Knowledge of Mother to Child Transmission of HIV and its prevention among Nigerian Women

We are grateful to the Academic Editor and Reviewers for reading through our manuscript and for providing valuable comments. We have revised the manuscript closely following the comments from the Reviewers. Below, we highlight how we have taken care of all the issues raised.

Reviewer #1: To me the work is worthwhile to raise awareness of the need to disseminate health and hygiene knowledge among women in order to raise their awareness and prevent new infections. I consider the efforts made by the authors to identify socio-economic determinants that can be the subject of targeted public health interventions to enhance women's awareness of the subject particularly interesting. 

Author: We appreciate the comments.

Reviewer: It is necessary to make a timely review of the spacing, in particular in the "results" section. I am unable to pronounce on the correctness of the statistical analyzes, as they are beyond my knowledge. However, considering the point of view of a reader who is on average fasting in statistical analysis, I believe that applied statistical analysis is expressed in a somewhat verbose way, and not free from some repetitions, ending up making the reading of this section not very smooth. 

Author: We appreciate the point of view expressed on the statistical section. We have read through all parts of the article and improved on the presentations of the statistical section. In particular, we have cut down aspects we feel could sound as repetition and tried to make the section as simple as possible.

Reviewer: Overall, I think the work is interesting and worthy of being brought to the attention of the public. The data exposed concern many women, thus acquiring considerable weight. The conclusions are in line with the findings and expressed with the necessary emphasis.

Author: We thank you for finding our work worthy of being brought to the public attention.

Reviewer #2: The article is interesting even if very complicated if you do not have great statistical skills. The methods and or results seem exhaustive. The data in table number 1 should be better specified, in which the characteristics of the general population are described (and not only of the women who then actually answered the survey of the five questions): it is not clear whether it refers to the total of 40,427 women.

Author: We appreciate the comments and have tried to improve on all aspects of the manuscript. We regret the mix up in the figures earlier presented in Table 1. We have now corrected everything. They are based on the number of women included in the analysis.

---

## [Editor Report · Decision Letter 1]

11 Jun 2021

Spatial Pattern and Determinants of Sufficient Knowledge of Mother to Child Transmission of HIV and its prevention among Nigerian Women

PONE-D-21-02315R1

Dear Dr. Olopha,

We’re pleased to inform you that your manuscript has been judged scientifically suitable for publication and will be formally accepted for publication once it meets all outstanding technical requirements.

Kind regards,

Sara Ornaghi, M.D., Ph.D.

Academic Editor

PLOS ONE
---

## [Editor Report · Acceptance letter]

18 Jun 2021

PONE-D-21-02315R1 

Spatial Pattern and Determinants of Sufficient Knowledge of Mother to Child Transmission of HIV and its prevention among Nigerian Women 

Dear Dr. Olopha:

I'm pleased to inform you that your manuscript has been deemed suitable for publication in PLOS ONE. Congratulations! Your manuscript is now with our production department. 

Kind regards, 

on behalf of

Dr. Sara Ornaghi 

Academic Editor

PLOS ONE